# The Role of Metal Nanoparticles in the Pathogenesis of Stone Formation

**DOI:** 10.3390/ijms25179609

**Published:** 2024-09-05

**Authors:** Varvara Labis, Igor Gaiduk, Ernest Bazikyan, Dmitry Khmelenin, Olga Zhigalina, Irina Dyachkova, Denis Zolotov, Victor Asadchikov, Ivan Kravtsov, Nikita Polyakov, Andrey Solovyev, Kirill Prusakov, Dmitry Basmanov, Ivan G. Kozlov

**Affiliations:** 1Department of Surgical Dentistry and Implantology of the N.D. Yushchuk Institute of Continuing Professional Education, Russian University of Medicine of the Ministry of Health of the Russian Federation, 4 Dolgorukovskaya St., 127006 Moscow, Russia; g21031969@yandex.ru (I.G.); prof.bazikian@gmail.com (E.B.); 2National Research Centre “Kurchatov Institute”, 59 Leninskiy Prospekt, 119333 Moscow, Russia; xorrunn@gmail.com (D.K.); zhigal@crys.ras.ru (O.Z.); zolotovden@crys.ras.ru (D.Z.); asad@crys.ras.ru (V.A.); 3N.F. Gamaleya Federal Research Center for Epidemiology & Microbiology, 18 Gamalei St., 123098 Moscow, Russia; 995662@gmail.com (I.K.); polyakovnb@gmail.com (N.P.); dronnias@gmail.com (A.S.); 4Vernadsky Institute of Geochemistry and Analytical Chemistry of Russian Academy of Sciences, 19 Kosygina St., 119991 Moscow, Russia; 5Lopukhin Federal Research and Clinical Center of Physical-Chemical Medicine of Federal Medical Biological Agency, 1A Malaya Pirogovskaya St., 119435 Moscow, Russia; kaprusakov@gmail.com (K.P.); basmanov.dmitry@gmail.com (D.B.); 6Institute of Professional Education, I.M. Sechenov First Moscow State Medical University, 8-2 Trubetskaya St., 119991 Moscow, Russia; immunopharmacology@yandex.ru

**Keywords:** metal nanoparticles, salivary gland stones, crystallization centers, immunoglobulin, endogenous infections, pathogenesis

## Abstract

The process of stone formation in the human body remains incompletely understood, which requires clinical and laboratory studies and the formulation of a new endogenous, nanotechnological concept of the mechanism of origin and formation of crystallization centers. Previously, the mechanism of sialolithiasis was considered a congenital disease associated with the pathology of the ducts in the structure of the glands themselves. To date, such morphological changes of congenital nature can be considered from the position of the intrauterine formation of endogenous bacterial infections complicated by the migration of antigenic structures initiating the formation of crystallization centers. The present work is devoted to the study of the morphology and composition of stones obtained as a result of surgical interventions for sialolithiasis. Presumably, nanoparticles of metals and other chemical compounds can be structural components of crystallization centers or incorporated into the conditions of chronic endogenous inflammation and the composition of antigenic structures, in complexes with protein and bacterial components. X-ray microtomography, X-ray fluorescence analysis, scanning transmission electron microscopy and microanalysis, mass spectrometry, and Raman spectroscopy were used to study the pathogenesis of stone formation. Immunoglobulins (Igs) of classes A and G, as well as nanoparticles of metals Pb, Fe, Cr, and Mo, were found in the internal structure of the stones. The complex of antigenic structures was an ovoid calcified layered matrix of polyvid microbial biofilms, with the inclusion of metal nanoparticles and chemical elements, as well as immunoglobulins. The obtained results of clinical and laboratory studies allow us to broaden the view on the pathogenesis of stone formation and suggest that the occurrence of the calcification of antigenic structures may be associated with the formation of IgG4-associated disease.

## 1. Introduction

To date, the pathogenesis of stone formation, in particular sialolithiasis, is considered a chronic immune-mediated inflammatory process and is a manifestation of chronic sialodenitis [1,2]. It is known that Mikulicz disease and syndrome, being manifestations of autoimmune pathology in the oral cavity, are associated with a lymphoid overgrowth of lacrimal and salivary gland tissue and, among other things, they belong to IgG4-associated diseases [3].

Microbiological aspects in the pathogenesis of stone formation are considered the main causes of stone formation in the salivary gland ducts. The formation of microbial biofilms with the inclusion of immunocompetent cells and calcium particles is a structural part of stones [4,5,6]. Therefore, the study of microbial composition in the formation of salivary stones is an urgent task in the study of the pathogenesis of sialolithiasis [7].

The study of the structure of salivary gland stones by transmission electron microscopy allowed us to divide them into three groups: calcified (CAL), organo-lipid (LIP), and mixed (MIX) [8]. When studying a large sample (44 units) of salivary stones by scanning electron microscopy and Raman spectroscopy, a layered structure of salivary stones was observed in 75% (33 units) of cases, and a homogeneous structure was observed in 25% (11 units) [9].

In 2022, Chinese scientists studied the crystal structure of submandibular salivary gland stones [10]. According to the criterion of a maximum transverse diameter ≥3 mm, a total of five submandibular stones meeting this requirement were included in the study. After washing the surface of the stone specimens, they were cut along the maximum transverse diameter. Scanning electron microscopy (SEM), energy dispersive X-ray spectroscopy (EDS) and polycrystalline X-ray diffraction (XRD) were used in the study to analyze the composition and structure of submandibular stones. The investigations revealed the presence of organic and inorganic compounds arranged evenly or unevenly. Hydroxyapatite (HAP) was the main inorganic component. Also, submandibular stones were saturated with carbon (C), oxygen (O), calcium (Ca) and phosphorus (P). In addition, the precursor of HAP, namely amorphous calcium phosphate (ACP), was detected. The tetrahedral substitution index (TSI) and Ca/P ratio reflected the degree of structural variability of the HAP crystal, which ranged from 5.62 to 90.71 and 1.10 to 1.35, respectively. As a result, it was concluded that the development of submandibular stones was influenced by chemical and structural differences of inorganic crystals, as well as the interaction of organics with inorganic components. Isomorphic substitution was accompanied by the appearance of inorganic crystals, resulting in changes in crystal structure. Organics can influence the appearance, aggregation and mineralization of HAP during its formation [10].

In addition, the study of the protein structure of salivary gland stones is also relevant. In 2021 Czaplewska et al. performed a qualitative analysis of stones using filter-assisted sample preparation (FASP) [11]. Fragmentation spectra analysis based on a human database identified 254 human proteins present in the stone structure. A multi-round bacterial-based search in the PEAKS Studio program allowed for the identification of 393 proteins of bacterial origin present in the extract obtained from sialolite, which has not been performed so far for this biological material [11]. The application of the SWATH-MS method allowed for a relative quantitative analysis of the human proteins present in the sediment. The results obtained agree with the classification of sialoliths proposed by Tretiakow [8]. The analysis allowed us, for the first time, to identify proteins whose levels differ in the samples under study, which may indicate the role of these proteins in the calcification process in different types of sialoliths [11].

The formation of crystallization centers and their structures are the basis of understanding the pathogenesis of sialolithiasis. In 2023, Korean scientists analyzed the dynamics of the formation of submandibular salivary gland stones [12]. The study aimed to identify the biological components of sialoliths, which have different ultrastructures and chemical compositions compared to choleliths and uroliths. Twenty-two specimens from 20 patients were obtained, examined histologically and analyzed by micro-CT, SEM, energy dispersive X-ray spectroscopy (EDS) and transmission electron microscopy (TEM). All sialoliths observed in this study had a central focus that was filled with an organoid matrix with an admixture of exosomal vesicles, loose calcium apatite crystals, and multiple bacteria. Micro-CT and SEM data clearly identified one or more central foci surrounded by a significantly calcified compact zone. The circular compact zone had band calcification with a thickness of about 1–3 mm and was usually located between the central focus and the peripheral multilayered zone. However, in some sialoliths (*n* = 5), there was a marked erosion of the compact zone due to the expansion of the multilayered zone, depending on the level of calcification and inflammation in the sialolith. When observing TEM images, many exosomal vesicles and degraded cytoplasmic organelles were found in the central focus, as well as some epithelial cells in the calcified matrix of the peripheral multilayered zone. In particular, EDS analysis showed the highest Ca/P ratio in the intermediate compact zone (1.77), followed by the central region of the nidus (1.39) and the peripheral multilayer zone (0.87). Taken together, these data suggest that the central focus, containing many inflammatory exosomes and degraded cytoplasmic organelles, has the potential to induce the band calcification of the compact zone followed by the multilayered deposition of detached salivary epithelial cells and salivary materials. Thus, the calcium apatite-based sialolith gradually increases in size and eventually impedes saliva flow and provides a site for bacterial infection [12].

The role of metal nanoparticles in stone formation remains unexplored. The field of nanotechnology has expanded over the past two decades and has made the transition from research to practical applications. Particles in the nanoscale range have become promising tools with wide applications in drug delivery, diagnostics, cosmetics and several other biological and non-biological fields [13]. These advances lead to questions about the safety of nanoparticles. Despite numerous studies on the toxicity and biocompatibility of nanoparticles, many of these questions remain unanswered. Nevertheless, these efforts have identified several approaches to minimize and prevent nanoparticle toxicity to promote safer nanotechnologies [14].

## 2. Results

### 2.1. The Results of X-ray Studies of Salivary Gland Stones

The results of micro-CT measurements of five (*n* = 5) salivary gland stone samples are presented in Figure 1. The linear absorption coefficient values in mm^−1^ are indicated on the scale.

The results of the study of the elemental composition of the surfaces of five samples of salivary gland concrements, conducted using X-ray fluorescence (XRF) analysis, are presented in Figure 2. Deciphering the obtained spectra revealed the presence in the surface layers of the studied samples, we found P, Ca (basic), Fe, Zn, Se, Br, and Sr. The presence of an Ar peak on all spectra is associated with measurements in air. The presence of an Ni peak is due to metal parts in the diffractometer design. There are insignificant differences in the elemental composition of the concrements, associated with different concentrations of the detected elements in the studied samples (peak height on the spectra) but not with the composition itself.

In order to study the elemental composition of the central region, one of the samples (sample No.4) was cut along the center (Figure 3a). X-ray fluorescence spectra were recorded on two obtained splits of the central part of the studied sample (Figure 3b(1,2)). The results of analysis of the measured spectra showed the presence of Pb in the central part. The spectrum from a control sample of pure Pb is also presented for comparison (Figure 3b(3)).

### 2.2. Results of Scanning Transmission Electron Microscopy (STEM) and Microanalysis

Salivary gland concretions were examined for the presence of metal-containing particles using STEM and microanalysis methods. It was found that the formations consist mainly of Ca, also containing chemical elements such as Mg, Na, O, F and P (Figure 4). C and Cu are present in the substrate and copper mesh.

In one of the samples, it was possible to detect a cluster of particles 10–50 nm in size containing metals such as Pb, Fe, Cr and Mo (Figure 5).

### 2.3. Results of Raman Spectroscopy Studies

To investigate the phase composition, Raman spectra from the internal regions of salivary gland stones were measured. Figure 6 shows the Raman spectrum from sample No.4 (Figure 1), which is typical of all stone samples studied. In terms of phase composition, they are identical. This agrees well with the fluorescence (XRF) data, from which it follows that the elemental composition is also the same. On the Raman spectrum, we detected the most-intense peak (band) at 962 cm^−1^ and weak peaks at 590 cm^−1^ and 1045 cm^−1^, which indicate the presence of a hydroxyapatite phase (Figure 6). For comparison, the Raman spectrum of pure hydroxyapatite from the database for minerals (RRUFF ID: R100225 [15]) is given.

In addition, a phenylalanine phase corresponding to the peak at 1005 cm^−1^ was detected [16]. According to the authors of the article, the detected peak refers to the stretching vibrations of the aromatic ring of phenylalanine, which is one of the characteristic side chains in the spectrum of collagen. In our opinion, this protein is a constituent component not only of collagen structures; rather, it correlates its presence in concretions with the phenomena of adhesive properties between different types of antigens.

### 2.4. Liquid Chromatography-Mass Spectrometry (LC-MS/MS) Analysis Results

A proteomic analysis of three stones removed from submandibular salivary glands was performed on a TimsTOF mass spectrometer (Bruker Daltonics, Bremen, Germany) in DDA mode, and 20 precursor ions per cycle (TOP 20) were fragmented. Proteolysis during sample preparation was performed using cyanogen bromide and trypsin. Expectedly, proteins belonging not only to the human proteome but also to other species, in particular those that make up the oral microbiome, were identified in the samples (Figure 7).

A search for peptides in the human protein database reliably identified between 17 and 88 proteins depending on the enzyme and sample used, with 14 proteins present in all three stone samples (Table 1).

A search in the UniProt database for data on the known functions of these proteins and their realization in the human body, in which they take part, has shown that, in addition to being involved in processes related to the normal functioning of the connective tissue of teeth and saliva, a number of proteins participate in various mechanisms of innate and acquired immune response. It should be noted that these proteins are involved in proinflammatory reactions of the immune response and regulate the involvement of the cellular component in local reactions of local immunity of mucous membranes. The complete list of all identified proteins and statistical parameters is given in Appendix A.

Given the significant proportion of proteins involved in immunological reactions, an additional search for peptide identification was performed using the human immunoglobulin database, resulting in peptides that reliably indicated the presence of immunoglobulins in all three samples. To confirm these findings, quantitative enzyme-linked immunosorbent assay (ELISA) analysis was performed for IgG, serum IgA, and secreted IgA immunoglobulins in samples obtained from the investigated submandibular salivary gland stones, normalized for the total protein obtained from the samples (Table 2).

Previously, the proteomes of submandibular salivary gland stones were analyzed in the works of other authors. Thus, in [17], the spectrum of the main proteins of sialoliths, regardless of their spectroscopic characteristics (lipid, calcified, or mixed), generally corresponded to the data obtained by us.

## 3. Discussion

To date, the development of most diseases has been correlated with genetic predisposition to their occurrence [18,19,20]. Consequently, we assume that in the process of embryogenesis in the intrauterine state, there is a circulation of the endogenous microflora of the mother through the bloodstream (umbilical vein) into the vascular bed of the child, or “transplacental colonization of the endogenous microbiome” (TCEM), due to planktonic forms and biofilms in the aggregate state. Such migratory colonization with the possibility of conditions for the initiation of acute and chronic inflammation in the placenta, emerging tissues and organs may be potentiating for the formation of intrauterine morphological changes in the child. This infestation can occur through the placental barrier, directly entering the bloodstream from mother to child [21,22,23,24,25]. This explains the congenital dilatation of ducts in the glands, with the early appearance of concrements in tissues and organs in children. This hypothesis requires interdisciplinary studies with the genomic sequencing of amniotic fluid, blood isolated from the umbilical vein, maternal endometrium, and other microbiologic niches.

The presence of metallic nanoparticles, including lead in the structure of stones, may be due to the ability to migrate, which we have shown earlier both in experimental modeling on a mouse model and in the structure of gallstones [14,26]. The presence of Pb, Fe, Cr, and Mo as components of the central points of crystallization indicates the ability of xenobiotics to enter tissues and organs, their migration, and their participation in pathological processes as structural components of the external environment. These trace elements can be a component of tobacco products, food additives and medicines [27,28]. If earlier the participation of metal ions in the structure of stones was discussed, today, we see the possibility of identifying metal nanoparticles in the composition of sialites. The problem of ecology and the consumption of products containing metal nanoparticles is reaching a new level of problems and requires further research.

We suggest that metal ions, nanoparticles and microparticles may be centers or participants of complex processes of complex antigenic structure formation in numerous forms: bacterial polyspecific biofilms formed by endogenous bacteria; metal ions; nanoparticles or microparticles migrating into the gland tissue as part of complexes with proteins in the form of immunoglobulins; and components of cellular structures of mucosal immunity [14]. The complex of metals and bacteria with secretory immunoglobulins included in the structure of the protein capsid shell can initiate stone formation, leading to the activation of mucosal immunity, which is not capable of eliminating it, and inducing matrix calcification as a protective reaction of the immune system based on the type of IgG4-associated reaction [29,30]. Consequently, a question arises concerning the possibility of considering the pathogenesis of sialolithiasis as a result of chronic autoinflammatory endogenous disease associated with a decrease in the function of gland secretion, which is caused by the impaired elimination of the complex of antigenic structures by means of reactions of innate and adaptive immunity, as an IgG4-associated disease [29].

## 4. Materials and Methods

### 4.1. Clinical Case

Patients with clinical signs of sialolithiasis were referred to the Department of Propaedeutics of Surgical Dentistry, Faculty of Dentistry, A.I. Evdokimov Scientific and Educational Institute (SEI) of Dentistry, and the Department of Surgical Dentistry and Implantology, SEI of Continuing Professional Education N.D. Yushchuk, “Russian University of Medicine”, of the Ministry of Health of Russia. The present study included five patients with concrements located in the ducts of the submandibular salivary glands. Figure 8 shows a clinical case of sialolithiasis of the submandibular salivary gland on the left side, characterized by the location of a concretion in the mouth of the gland duct. Patient K (44 years old, male sex) was admitted with complaints of pain in the hyoid region on the left side (Figure 8), as well as redness, swelling, and inability to eat. Figure 9 shows the image of cone-beam computed-tomography (CBCT) slices. Figure 10 shows the stage of surgery for the removal of a nodule from the duct of the submandibular salivary gland on the left.

### 4.2. X-ray Micro-CT

Microtomographic studies of five samples of salivary gland concrements removed from different patients were performed on a TOMAS X-ray microtomograph. In these measurements, a monochromator crystal made of pyrolytic graphite was used (the beam size on the object is about 2 cm). The use of monochromatic radiation allows for measuring the real value of the linear X-ray attenuation coefficient, which is important when studying the elemental composition of samples. The measurements were carried out using X-ray tubes with molybdenum and silver anodes (energies of 17.5 and 22.2 keV, respectively). When these energies are used, the sample under study is, on the one hand, transparent enough to maintain a high signal-to-noise ratio in the images, and on the other hand, the boundaries of strongly absorbing internal parts are clearly visible.

The geometry of the experiments was as follows: a source-sample distance of 1.2 m and a sample-detector distance of 0.02 m. With such geometry in the experiment, the divergence of radiation is negligibly small, which is important for image analysis. The probing conditions were as follows: an accelerating voltage of 40 kV and a current of 40 mA. For tomographic studies, the samples were rotated relative to a fixed vertical axis, and 400 projections were measured with a step of 0.5 degrees and exposure times of 5 s (molybdenum anode) and 8 s (silver anode). An XIMEA-xiRay11 high-resolution X-ray detector was used for the measurements, which produces 9 µm resolution images with a field of view of 36 mm × 24 mm. Reconstruction was performed using the algebraic method.

### 4.3. X-ray Fluorescence Analysis (XRF)

The elemental composition of salivary gland concrements samples was determined by X-ray fluorescence analysis on a “DITOM-M” X-ray diffractometer using a detector-spectrometer X-123SDD (Amptek, Bedford, MA, USA). The source of radiation was an X-ray tube with a silver anode, the radiation of which was monochromatized by means of a single reflection from a highly perfect, symmetric silicon crystal with an orientation of (111). The emission energy corresponded to the characteristic line Kα1 of silver and was 22.162 keV. The lower limit of measurements was limited by the parameters of the sensitive element of the detector and was ~1 keV. The energy resolution of the fluorescence emission spectra taken from the sample under study was ~160 eV. The geometry of the experiments was as follows: the source-sample distance was about 1 m and the sample-detector distance was about 0.02 m. The size of the probing beam on the sample was ~1 mm horizontally and ~5 mm vertically. The probing conditions were as follows: an accelerating voltage of 40 kV and a current of 40 mA. The exposure time was 20 min.

### 4.4. Scanning Transmission Electron Microscopy (STEM) and Microanalysis

Salivary gland concrements were studied on an Osiris transmission electron microscope at an accelerating voltage of 200 kV. The microscope was equipped with a special system of detectors that allowed for obtaining large-area distribution maps of chemical elements in a few minutes. Samples were prepared by grinding to powder in an agate mortar and then dispersed in an ultrasonic bath in acetone for several minutes, after which they were applied to copper grids with a thin amorphous substrate.

### 4.5. Raman Spectroscopy

Raman measurements were conducted at room temperature using an INTEGRA Spectra confocal Raman microscope (NT-MDT, Zelenograd, Moscow, Russia) with a sample scanning-stage system. The spectra were acquired using diffraction grating (600 lines per mm) and recorded with an electron-multiplying charge-coupled device (Newton, Andor; Abingdon, Oxfordshire, UK) cooled down to −65 °C. The grating was set up so that the desired range of wavelengths (500–1300 cm^−1^) reached the detector. The grid position was calibrated using a neon lamp. For the measurements, a 785 nm laser was used; the spectra were recorded using a 100 × 0.9 NA Olympus objective with a pinhole size of 100 µm. Laser radiation was focused into a minimum spot on the sample surface.

Samples for the study were prepared as follows. The initial stones were placed in an Eppendorf, then poured with epoxy resin. After curing, they were cut in the central part of the stone and further polished to obtain a smooth and flat surface.

### 4.6. Mass Spectrometry

#### 4.6.1. Treatment of Samples with Cyanogene Bromide

First, 70 µL of 100% formic acid was added to samples of salivary gland concrements and sonicated in an ultrasonic bath for 15 min. To the solution, 30 µL of water and excess bromocyanine were added, and samples were left in a dark place at room temperature for at least 12 h. After completion, the reaction mixture was diluted to 200 μL with bidistilled water and dried to dryness using a Savant SPD121P vacuum concentrator (Thermo Fisher Scientific, Waltham, MA, USA). The resulting precipitate was dissolved in 20 μL of 0.1% formic acid. The sample was desalted on in-house columns with reversed-phase membranes (analog C18), and the sample was eluted with 90% solution of acetonitrile in water. The eluate was dried to dryness, and the resulting precipitate was dissolved in 20 μL of 0.1% formic acid. The sample was used for liquid chromatography-mass spectrometry (LC-MS/MS analysis).

#### 4.6.2. Treatment of Samples with Trypsin

From 70 to 100 µL, 100% formic acid was added to samples of salivary gland concrements and sonicated in an ultrasonic bath for 15 min. To the solution, we added 30 µL of water. The resulting solution was neutralized with aqueous ammonia solution. To the obtained reaction mixture, we added 1 μL of 200 mM TCEP solution in 50 mM ABC, 1 μL of 100 mM CAA solution in 50 mM ABC, and 2 μL of trypsin (mass-spectrometry grade, Promega Corporation, Madison, USA) in 50 mM acetic acid (concentration 1 mg/mL), which we incubated at 370 °C for 12 h. At the end of the reaction, 2.5 μL of trifluoroacetic acid (TFA) was added to the solution. The sample was desalted on homemade columns as described above.

#### 4.6.3. Liquid Chromatography-Mass Spectrometry (LC-MS/MS Analysis)

LC-MS/MS analysis was performed using a NanoElute ultrahigh-pressure nanoflow system (Bruker Daltonics, Bremen, Germany) coupled to a TimsTOF mass spectrometer (Bruker Daltonics, Bremen, Germany). A CaptiveSpray ion source (Bruker Daltonics, Bremen, Germany) was used. The sample was loaded onto a BrukerTEN capillary column with a C18 phase (10 cm length, 75 μm inner diameter, 1.9 μm particle size, 120 Å pore size; IonOptiks, Fitzroy, Australia). Peptides were separated at 50 °C using a gradient for 60 min at a flow rate of 300 nl/min (mobile phase A: 0.1% formic acid in water; mobile phase B: 0.1% formic acid in acetonitrile). We use a step gradient as follows: 2–35% phase B for 60 min, 35–95% phase B from 60.0 to 60.5 min, and 95% phase B from 60.5 to 66.2 min, and the total run time of the assay was 66.1 min.

The TimsTOF mass spectrometer was operated in the ion-mobility off mode. Mass spectra for MS and MS/MS scans were recorded in the range of 100–1750 m/z. Data acquisition was performed using a 3 s cycle at 8 Hz. An active exclusion time of 0.4 min was applied to precursors reaching 5000 intensity units.

#### 4.6.4. Mass Spectrometry Data Processing

The mass spectra obtained were processed using PEAKS Studio 11 software (Bioinformatics Solution Inc., Waterloo, ON, Canada), using a database containing the Uniprot database sequence as well as known contaminants. Additionally, the human immunoglobulin database was searched. The following parameters were used in the search: a parent ion mass-determination error of 15 ppm, fragment ions of 0.02 Da, trypsin enzyme, a semi-specific cleavage specificity, a possible post-stranding modification of methionine oxidation, and a maximum of three post-translational modifications (PTM) per peptide. The plausibility criterion for peptides was 0.1% FDR, and for proteins, it was 1% FDR. The number of identified peptides per protein was greater than or equal to two. Unipept desktop Version 2.0.1 [31] was used to visualize the taxonomic affiliation of proteins.

The mass spectrometry proteomics data have been deposited to the ProteomeXchange Consortium via the PRIDE [32] partner repository with the dataset identifier PXD054153.

### 4.7. Determination of Immunoglobulin Concentrations by Enzyme-Linked Immunosorbent Assay (ELISA) Method

Samples of salivary gland concrements in phosphate-salt buffer with a mixture of protease inhibitors (Halt™ Protease and Phosphatase Inhibitor Cocktail, Thermo Fisher Scientific, Waltham, MA, USA) were placed for 30 min in an ultrasonic bath (UNITRA UM-05 60 W). The sample tubes were then centrifuged at 10,000 g for 10 min, and the supernatant was collected and transferred to a new microtube.

The protein concentration of each sample was determined fluorometrically (Qubit™ Protein and Protein Broad Range (BR) Assay Kit).

Commercial kits (A-8662 IgG total-IFA-BEST, A-8668 IgA secretory-IFA-BEST, A-8666 IgA total-IFA-BEST, VectorBest, Moscow, Russia) were used to determine the concentration of immunoglobulins (type G, A total and A secretory) in the obtained solution. The assay was performed according to the kit manufacturer’s instructions. The samples were pre-diluted four-fold with binding solutions from the respective kits.

## 5. Conclusions

As a result of the conducted studies of the role of metal nanoparticles as crystallization centers in the structure of the resulting sialoliths, elements such as lead (Pb), iron (Fe), chromium (Cr), and molybdenum (Mo) were identified. These chemical elements, determined at the nanoscale, correlate with foreign structures for the immune cells of the mucous membrane. The ability of these particles to migrate and be excreted from the body, due to the secretory function of the salivary glands, makes them participants in the formation of a complex of antigenic determinants during stone formation. STEM analysis during the grinding of sialoliths revealed the presence of metal nanoparticles in the structures of the studied samples. It is important to note that the content of foreign metal inclusions directly in the central part of the sialoliths is confirmed by X-ray fluorescence. This aspect became the basis for studying the location of metal nanoparticles in relation to the protein components of stones. The method of X-ray microtomography made it possible to identify areas with low density. The low-density sites are likely associated with the detected protein components identified by mass spectrometry. It should be noted that between 17 and 88 proteins were detected, while 14 proteins were present in all the studied samples.

In the study of submandibular salivary gland concrements, it was shown that the content of oral mycobiome proteins was not only polyspecies bacteria and eukaryotes but also direct protein structures of immune system elements—in particular, serum and secretory immunoglobulins IgA and IgG—which establishes the very process of stone formation as immune-mediated. Based on the obtained results of the studies, it can be stated that the process of sialolithiasis is an endogenous infection associated with the formation of a multicomponent complex of antigenic determinants. It is based on the organic component in the form of proteins of polyspecific bacteria, metal particles of different sizes and elemental compositions, antibodies of IgA and IgG classes, and inorganic components (Ca, P, etc.), which are a protective reaction of the immune system in the form of the calcification of the organic matrix. In our opinion, the inability to eliminate the complex of antigenic structures leads to the formation of calcifications, which is a manifestation of IgG4-associated diseases. This assumption requires further studies.

## Figures and Tables

**Figure 1 ijms-25-09609-f001:**
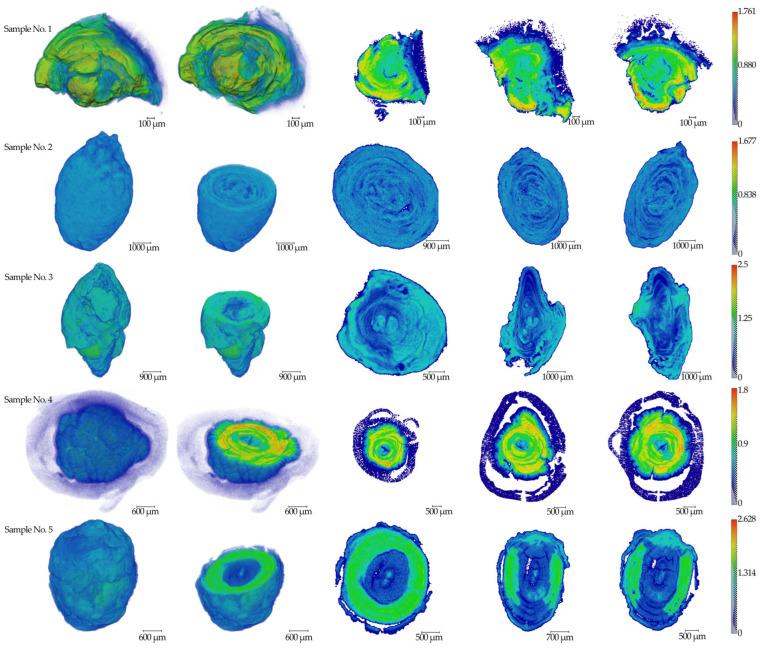
The results of tomographic reconstructions of samples No.1–5. Three-dimensional images and three mutually perpendicular sections from the center parts are shown for all specimens. The color scale of the linear absorption coefficient in mm^−1^ is presented on the right side.

**Figure 2 ijms-25-09609-f002:**
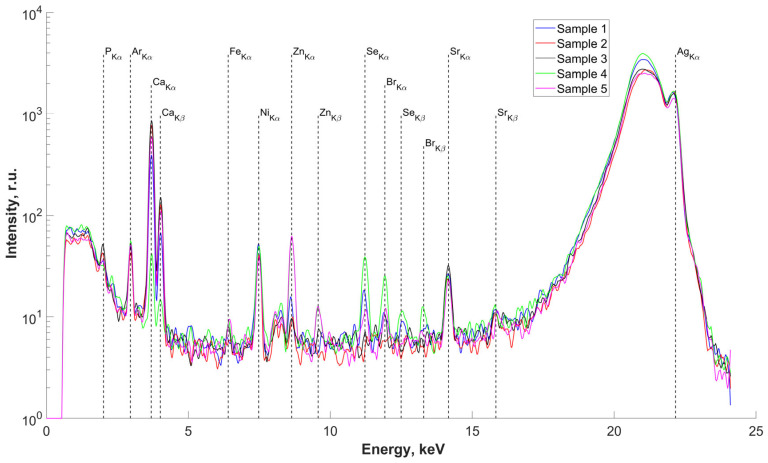
Fluorescence spectra of samples No.1–5. Energy 22.162 keV.

**Figure 3 ijms-25-09609-f003:**
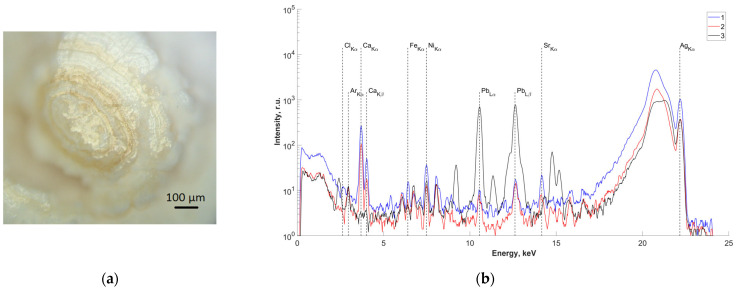
(**a**) Optical image of the surface of the studied slip of sample No.4; (**b**) fluorescence spectra of the central part of sample No.4 (energy 22.162 keV): 1,2—salivary gland stone; 3—lead sample.

**Figure 4 ijms-25-09609-f004:**
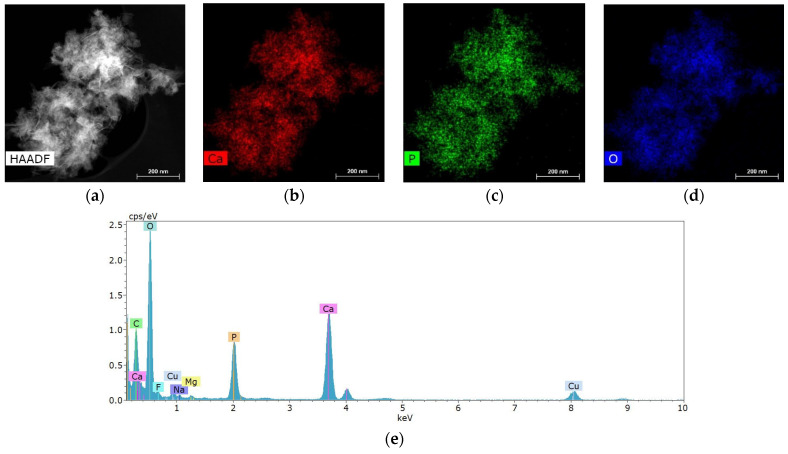
Characteristic structure and elemental composition of particles from salivary gland stones: (**a**) HAADF STEM image; (**b**–**d**) chemical element distribution maps; (**e**) energy dispersive spectrum.

**Figure 5 ijms-25-09609-f005:**
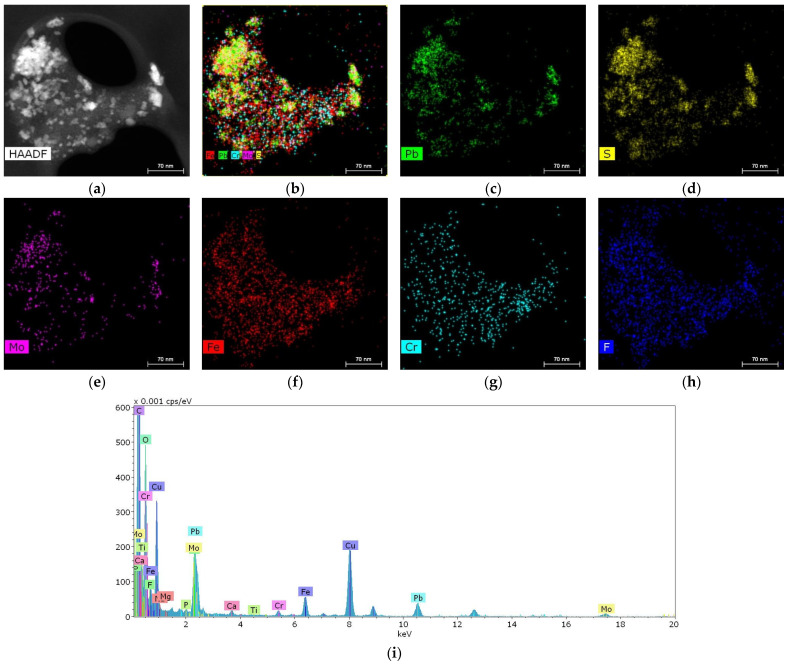
Structure and elemental composition of metal nanoparticles found in salivary gland stones: (**a**) HAADF STEM image; (**b**–**h**) chemical element distribution maps; (**i**) energy dispersive spectrum.

**Figure 6 ijms-25-09609-f006:**
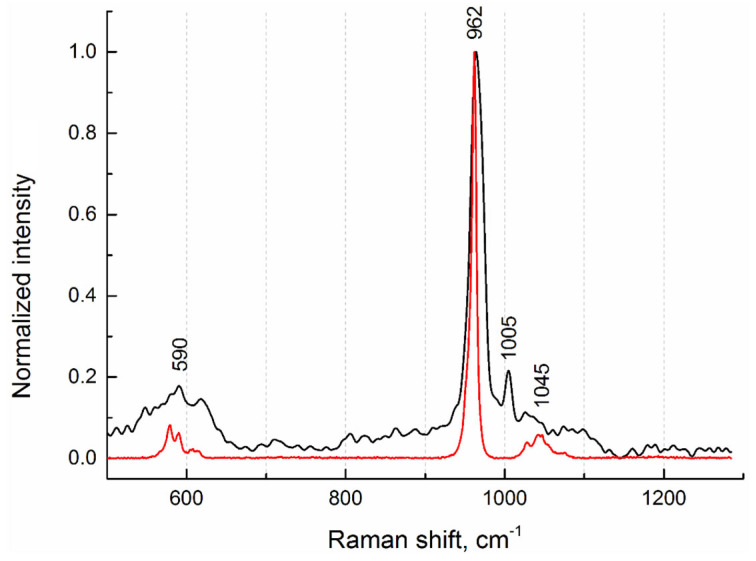
Raman spectra of sample No.4 (black curve) and hydroxyapatite (red curve) from the database (RRUFF ID: R100225).

**Figure 7 ijms-25-09609-f007:**
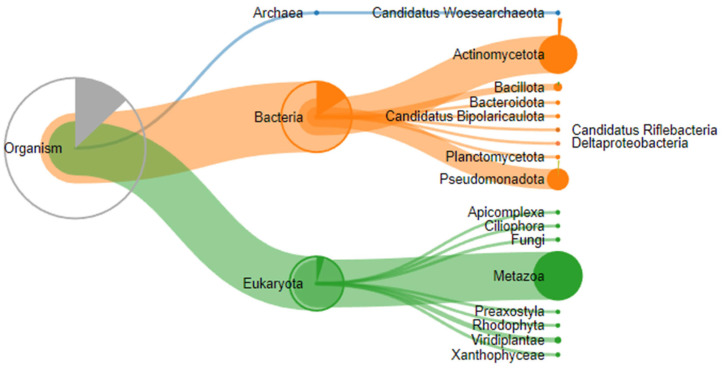
Taxonomic classification of proteins identified by proteomic analysis.

**Figure 8 ijms-25-09609-f008:**
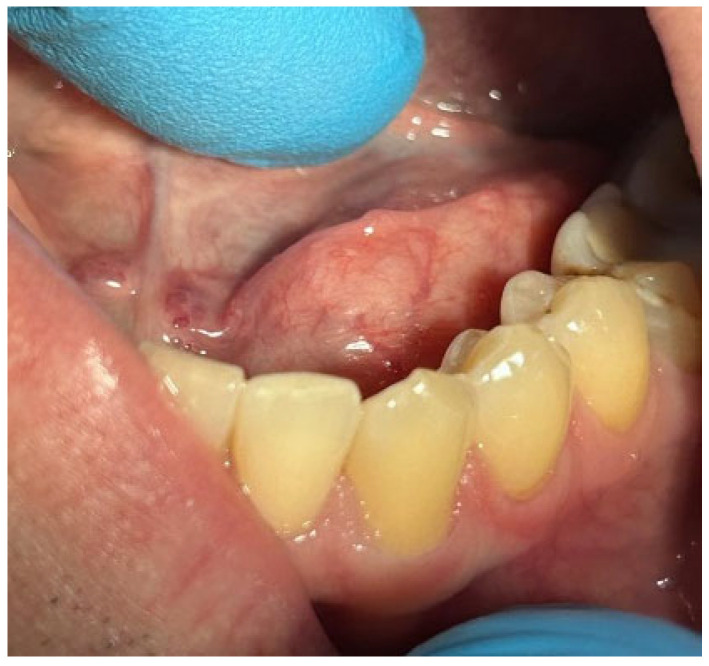
Condition of the mucous membrane of the hyoid region on the left side at the time of the patient’s treatment.

**Figure 9 ijms-25-09609-f009:**
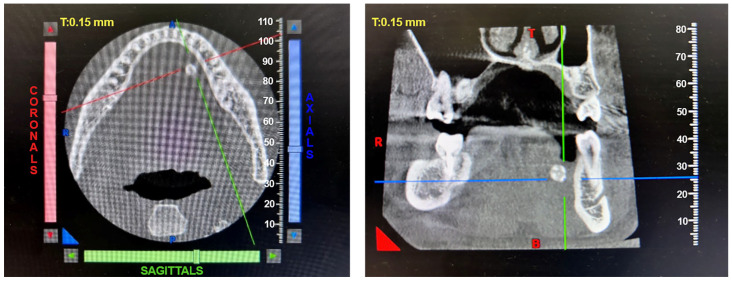
CBCT of patient K. (44 years old, male sex) before surgery.

**Figure 10 ijms-25-09609-f010:**
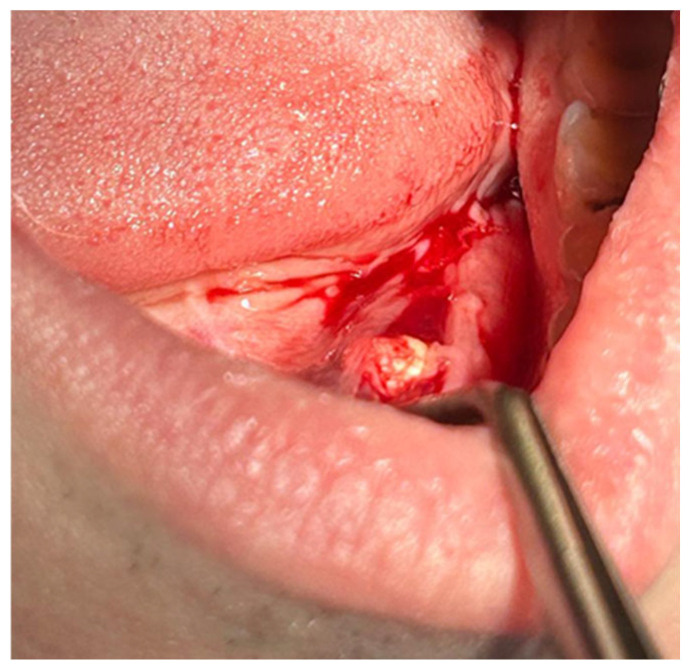
Surgery for removal of a concrement from the duct of the submandibular salivary gland on the left.

**Table 1 ijms-25-09609-t001:** Proteins detected in samples of salivary gland concretions.

Protein ID, Trypsin Cleavage	Protein ID, BrCN Cleavage	Protein Name	Function
**P02808|STAT_HUMAN**	P02808|STAT_HUMAN	Statherin	Salivary protein that stabilizes saliva supersaturated with calcium salts by inhibiting the precipitation of calcium phosphate salts. It also modulates hydroxyapatite crystal formation on the tooth surface.
**P02814|SMR3B_HUMAN**	P02814|SMR3B_HUMAN	Submaxillary gland androgen-regulated protein 3B	Predicted to be involved in cellular response to lipopolysaccharide; negative regulation of peptidase activity; and regulation of sensory perception of pain. Located in extracellular exosome.
**P06702|S10A9_HUMAN**	P06702|S10A9_HUMAN	Protein S100-A9 (Calgranulin-B)	Calcium- and zinc-binding protein which plays a prominent role in the regulation of inflammatory processes and immune response.
**P08311|CATG_HUMAN**	P08311|CATG_HUMAN	Cathepsin G	Serine protease with trypsin- and chymotrypsin-like specificity. Also displays antibacterial activity against Gram-negative and Gram-positive bacteria independent of its protease activity.
**P08493|MGP_HUMAN**	P08493|MGP_HUMAN	Matrix Gla protein	Associates with the organic matrix of bone and cartilage. Thought to act as an inhibitor of bone formation.
**P05109|S10A8_HUMAN**	P05109|S10A8_HUMAN	S100A8	Calcium- and zinc-binding protein which plays a prominent role in the regulation of inflammatory processes and immune response.
**P12724|ECP_HUMAN**	P12724|ECP_HUMAN	Eosinophil cationic protein	Cytotoxin and helminthotoxin with low-efficiency ribonuclease activity. Possesses a wide variety of biological activities.
**Q8NFU4|FDSCP_HUMAN**		Follicular dendritic cell secreted peptide	Secreted mediator acting upon B-cells.
**P15515|HIS1_HUMAN**		Histatin-1	Major precursors of the protective proteinaceous structure on tooth surfaces (enamel pellicle).
**P80511|S10AC_HUMAN**		Protein S100-A12	Calcium-, zinc- and copper-binding protein, which plays a prominent role in the regulation of inflammatory processes and immune response.
**P02810|PRPC_HUMAN**		Salivary acidic proline-rich phosphoprotein 1/2	Highly potent inhibitors of crystal growth of calcium phosphates. They provide a protective and reparative environment for dental enamel which is important for the integrity of the teeth.
	P61626|LYSC_HUMAN	Lysozyme C	Bacteriolytic function.
	P35527|K1C9_HUMAN	Keratin, type I cytoskeletal 9	May serve an important special function either in the mature palmar and plantar skin tissue or in the morphogenetic program of the formation of these tissues. Plays a role in keratin filament assembly.
	P24158|PRTN3_HUMAN	Myeloblastin	Serine protease that degrades elastin, fibronectin, laminin, vitronectin, and collagen types I, III, and IV; play a role in neutrophil transendothelial migration.
**Immunoglobulins**	Immunoglobulins	Immunoglobulins	Humoral immune response.

**Table 2 ijms-25-09609-t002:** Immunoglobulin content of salivary gland stones.

Immunoglobulin/Protein Total (Pure Mass Fraction) in Sample			
	IgG (µg/mg)	IgA (µg/mg)	Secretory IgA (µg/mL)
Sample 2	not available	not available	not available
Sample 3	not available	0.32	0.24
Sample 5	7.33	18.03	3.86

Both serum IgA and secretory IgA were detected in two of the three samples, with one of the samples also showing significant amounts of IgG. The absence of immunoglobulins in sample No.2 found by ELISA may be due to the detection limit of the ELISA kits used. Complete data with calibration curves and normalization data for total protein in the samples are presented in Appendix A.

## Data Availability

The data presented in this study are available on request from the corresponding author. The data are not publicly available because they are part of the patent application, they cannot be provided to a wide range of the professional community until it is received by the authors. In particular, the research results are part of the dissertation work that has not been defended. After the defense of the scientific qualification work, the data can be published in the public domain.

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
