# Peer review of "The Role of Metal Nanoparticles in the Pathogenesis of Stone Formation"

_ijms, 2024, doi:10.3390/ijms25179609_

Round 1

Reviewer 1 Report

Comments and Suggestions for Authors

the present work is devoted to the study of morphology and composition of stones obtained as a result of surgical interventions for sialolithiasis.

Based on the obtained research results, the authors claim that the process of sialolithiasis is an endogenous infection associated with the formation of a multicomponent complex of antigenic determinants based on the organic component in the form of proteins of polyspecific bacteria, metal particles of different size and elemental composition, antibodies of IgA and IgG class and inorganic components - Ca, P, etc., which are a protective reaction of the immune system in the form of calcification of the organic matrix. The authors argue that the inability to eliminate the complex of antigenic structures leads to the formation of calcifications, which is a manifestation of IgG4-associated diseases.

The topic is very novel and presents various study techniques, which adds significant value to the paper. However, it has a disorganized structure. Although it has a clear introduction and a well-developed body, the lack of coherence in the structure impacts the overall presentation of the work.

References should not be included in the conclusions; they should reflect the results and interpretations of the authors themselves. The conclusions are the section where the study's findings are synthesized, research questions are answered, and a critical reflection on the meaning of the results is provided. Including references in this section could dilute the originality of the analysis and divert attention from what the authors have directly found or interpreted.

The images are very representative and appropriate.

    4o

Reviewer 2 Report

Comments and Suggestions for Authors

This manuscript aims to elucidate the mechanisms underlying the formation of stones in the human body, focusing specifically on sialolithiasis. The authors introduce an endogenous, nanotechnological concept to explain the formation of crystallization centers. They identify that metals ions (Pb, Fe, Cr, Mo) and other chemical compounds are integral to these crystallization centers, or they become part of them through chronic endogenous inflammation. While the research broadens the understanding of stone pathogenesis and suggests a link between the calcification of antigenic structures and IgG4-related disease, several issues need addressing:

1. The authors conclude that “metal ions, nanoparticles and microparticles may be centers or participants of complex processes of complex antigenic structure formation in the form of bacterial polyspecific biofilms formed by endogenous bacteria.” However, the evidence for nanoparticles being the center of the stone is unclear. From Figure 14, nanoparticles appear randomly distributed. How did the authors confirm that these nanoparticles are central to stone formation?

2. In line 336, the authors note that the TEM sample preparation destroyed the stone's structure. It is recommended that they employ embedded slices for TEM to preserve the structural integrity of the samples.

3. The manuscript should ensure that the full name is accompanied by its abbreviation upon the first mention. For example, SEM is first mentioned in line 73 but is shown again in line 104 without an accompanying full name.

4. Figures 1-10 are results from similar experiments and should be combined into a single figure to streamline the presentation and improve clarity.

5. The meaning of the colors in Figures 1-10 should be explained in the figure legends to aid in the interpretation of the results.

6. The manuscript should include the percentage of each element identified in Figures 13 and 14 to provide a quantitative understanding of the composition of the stones.

Round 2

Reviewer 2 Report

Comments and Suggestions for Authors

The author has solved all issues.

Author Response

Comment 1: The conclusions section should reflect the results and interpretations of the
authors themselves. Please add your conclusions in this section. In your answer to the reviewer 1 you specified that ,,The Discussion and Conclusions section has been redesigned,, but there is no change in the conclusions section.

Answer 1: The Conclusion section has been added to the article.

Comment 2: we consider 19 authors  are too many and the contributions of the
authors should be reviewed and clarified. Please double check this remark and
adjust the authors and their contribution respecting the international
authorship criteria. If they were contributors to the study that don’t
fulfill these criteria please mention them in the Aknowledgements section."

Answer 2: The membership of the authors has not been changed. The uploaded file contains clarifying information about the contribution of each author to this work.